SpLitteR: diploid genome assembly using TELL-Seq linked-reads and assembly graphs

Tolstoganov Ivan 1
Chen Zhoutao 2
Pevzner Pavel 3
http://orcid.org/0000-0002-2937-9259 Korobeynikov Anton 4 5 anton@korobeynikov.info
1 Department of Mathematics, Science for Life Laboratory, Stockholm University , Stockholm , Sweden
2 Universal Sequencing Technology Corporation , Carlsbad, California , United States
3 Department of Computer Science and Engineering, University of California, San Diego , San Diego, California , United States
4 Department of Statistical Modelling, Saint Petersburg State University , Saint Petersburg , Russia
5 Institute of Applied Computer Science, ITMO University , Saint Petersburg , Russia
Oppert Brenda
Electronic publication date: 2024 Sep 27
Publication date: 2024
Volume: 12
Electronic Location ID: e18050
Received 2023 Nov 17; Accepted 2024 Aug 15
Copyright: © 2024 Tolstoganov et al.
Copyright year: 2024
Copyright holder: Tolstoganov et al.
License: This is an open access article distributed under the terms of the Creative Commons Attribution License, which permits unrestricted use, distribution, reproduction and adaptation in any medium and for any purpose provided that it is properly attributed. For attribution, the original author(s), title, publication source (PeerJ) and either DOI or URL of the article must be cited.
License URL: https://creativecommons.org/licenses/by/4.0/

Keywords: Tell-seq, Assembly graph, Repeat resolution

Funding: Russian Science Foundation 19-14-00172 This work was supported by the Russian Science Foundation (No. 19-14-00172 to Ivan Tolstoganov and Anton Korobeynikov). The funders had no role in study design, data collection and analysis, decision to publish, or preparation of the manuscript.

==============================
Background

Recent advances in long-read sequencing technologies enabled accurate and contiguous de novo assemblies of large genomes and metagenomes. However, even long and accurate high-fidelity (HiFi) reads do not resolve repeats that are longer than the read lengths. This limitation negatively affects the contiguity of diploid genome assemblies since two haplomes share many long identical regions. To generate the telomere-to-telomere assemblies of diploid genomes, biologists now construct their HiFi-based phased assemblies and use additional experimental technologies to transform them into more contiguous diploid assemblies. The barcoded linked-reads, generated using an inexpensive TELL-Seq technology, provide an attractive way to bridge unresolved repeats in phased assemblies of diploid genomes.

Results

We developed the SpLitteR tool for diploid genome assembly using linked-reads and assembly graphs and benchmarked it against state-of-the-art linked-read scaffolders ARKS and SLR-superscaffolder using human HG002 genome and sheep gut microbiome datasets. The benchmark showed that SpLitteR scaffolding results in 1.5-fold increase in NGA50 compared to the baseline LJA assembly and other scaffolders while introducing no additional misassemblies on the human dataset.

Conclusion

We developed the SpLitteR tool for assembly graph phasing and scaffolding using barcoded linked-reads. We benchmarked SpLitteR on assembly graphs produced by various long-read assemblers and have demonstrated that TELL-Seq reads facilitate phasing and scaffolding in these graphs. This benchmarking demonstrates that SpLitteR improves upon the state-of-the-art linked-read scaffolders in the accuracy and contiguity metrics. SpLitteR is implemented in C++ as a part of the freely available SPAdes package and is available at https://github.com/ablab/spades/releases/tag/splitter-preprint.

Introduction

The recently developed linked-read technologies, such as stLFR (McElwain et al., 2017), TELL-Seq (Chen et al., 2020), and LoopSeq (Callahan et al., 2021), are based on co-barcoding of short reads from the same long DNA fragment. They start with the distribution of long DNA fragments over a set of containers marked by a unique barcode. Afterward, long fragments within the containers are barcoded, sheared into shorter fragments, and sequenced. The resulting library consists of short reads marked by the barcode corresponding to the set of long fragments, or linked-reads. Portions of this text were previously published as part of a preprint (Tolstoganov et al., 2022).

Various tools, such as Athena (Bishara et al., 2018), cloudSPAdes (Tolstoganov et al., 2019), Supernova (Weisenfeld et al., 2017), and TuringAssembler (Chen et al., 2020), were developed to generate de novo genome assembly from linked-reads alone. However, even though linked-reads result in more contiguous assemblies than assemblies based on non-linked short reads, all these tools generate rather fragmented assemblies of large genomes and metagenomes (Rhie et al., 2021; Zhang et al., 2023). For large genomes and metagenomes, long high-fidelity (HiFi) reads proved to be useful in generating highly-accurate and contiguous assemblies (Nurk et al., 2020; Shafin et al., 2020; Kolmogorov et al., 2020; Cheng et al., 2021; Nurk et al., 2022; Rautiainen et al., 2023; Garg, 2021). Still, even though HiFi reads enabled the first complete assembly of the human genome by the Telomere-to-Telomere (T2T) consortium (Nurk et al., 2022), HiFi assemblies do not resolve some long repeats and thus are often scaffolded using supplementary technologies, such as Hi-C reads, Oxford Nanopore (ONT) ultralong reads, and Strand-seq reads (Nurk et al., 2022; Garg, 2023). Scaffolding methods based on inexpensive linked-reads represent a viable alternative to other supplementary technologies since they combine the low cost of short reads and the long-range information encoded by linked-reads originating from the same barcoded fragment.

Although the state-of-the-art linked-read scaffolders, such as Architect (Kuleshov, Snyder & Batzoglou, 2016), ARKS (Coombe et al., 2018), Physlr (Afshinfard et al., 2022), and SLR-superscaffolder (Guo et al., 2021) improve the contiguity of HiFi assemblies, they do not take advantage of the assembly graph and thus ignore the important connectivity information encoded by this graph. In addition, these tools are not applicable to diploid assemblies and complex metagenomes with many similar strains.

We present the SpLitteR tool that uses linked-reads to improve the contiguity of phased HiFi assemblies. In contrast to existing linked-reads scaffolders, it utilizes the assembly graph and was developed with diploid assemblies in mind. Given a linked-read library and a HiFi assembly graph in the GFA format, SpLitteR resolves repeats in the assembly graph using linked-reads and generates a simplified (more contiguous) assembly graph with corresponding scaffolds.

Materials and Methods

Figure 1 illustrates the SpLitteR workflow. First, SpLitteR maps the barcoded TELL-Seq reads to the edges of the assembly graph, identifies the uniquely mapped reads, and stores their barcodes for each edge (see “Aligning barcoded reads” for details). We assume that the genome defines an (unknown) genomic traversal of the assembly graph. A vertex in a graph is classified as branching if both its in-degree and out-degree exceed 1 (each branching vertex in the graph represents a genomic repeat). Given an incoming edge e into a branching vertex v, SpLitteR attempts to find an outgoing edge next(e) that immediately follows e in the genomic traversal by analyzing all linked reads that map to both the in-edge e and all out-edges from v (see “Repeat resolution”). A vertex is classified as resolved if SpLitteR finds a follow-up edge for each incoming edge into this vertex. SpLitteR further simplifies the assembly graph by splitting the resolved vertices in such a way that each matched pair of an in-edge and an out-edge is merged into a single edge, reducing the number of unphased haplotypes in case of a diploid assembly. Finally, it outputs the results of the repeat resolution procedure both as the set of scaffolds and as the simplified assembly graph. The repeat resolution procedure has both diploid and metagenomic modes.

Figure 1 Brief summary of the SpLitteR workflow.

In this toy example, the assembly graph is represented as the multiplex de Bruijn graph (Kolmogorov et al., 2020) where vertices are labeled by k-mers of varying sizes. Reads with the same barcode are represented by the same color (each barcode contains only two reads). Reads are mapped to the assembly graph based on their unique k-mers, i.e., k-mers which occur only once in the edges of the assembly graph (k = 3 for this toy example). Since yellow reads do not contain unique 3-mers, they remain unmapped. SpLitteR resolves vertices in the multiplex de Bruijn graph by assigning in-edges to their follow-up out-edges based on the barcode information.

In this toy example, the assembly graph is represented as the multiplex de Bruijn graph (Bankevich et al., 2022) where vertices are labeled by k-mers of varying sizes. Reads with the same barcode are represented by the same color (in Fig. 1, each barcode contains only two reads). Reads are mapped to the assembly graph based on their unique k-mers, i.e., k-mers which occur only once in the edges of the assembly graph (k = 3 for this toy example). In the middle subfigure, substrings of the edges to which the unique k-mers were mapped are colored in accordance with the color of the read containing the k-mers. Since yellow reads do not contain unique 3-mers, they remain unmapped. SpLitteR resolves vertices in the multiplex de Bruijn graph by assigning in-edges to their follow-up out-edges based on the barcode information.

Data preparation

To generate the TELL-Seq dataset from the HG002 genome, 5 ng high molecular weight genomic DNA, extracted from GM24385 (HG002) cells, was used to construct TELL-Seq WGS libraries based on the manufacturer’s user guide for large genome library preparation (available at https://universalsequencing.com/pages/library-prep-guides). These libraries were sequenced as 2 × 150 paired-end reads on a NovaSeq instrument.

A fecal sample of the SHEEP dataset was taken from a young (<1 year old) wether lamb of the Katahdin breed. DNA was extracted in small batches from approximately 0.5 g per batch using the QIAamp PowerFecal DNA Kit, as suggested by the manufacturer (Qiagen, Hilden, Germany), with moderate bead beating. A total of 5 ng DNA was used to construct TELL-Seq WGS libraries based on the manufacturer’s user guide for large genome library preparation (Universal Sequencing Technology). These libraries were sequenced as 2 × 150 paired end reads on a NovaSeq instrument.

Since homopolymer-compressed HiFi reads have significantly lower error rates than raw HiFi reads, assembly graphs produced by some of the existing HiFi-based assemblers, such as LJA (Bankevich et al., 2022), are homopolymer-compressed. Thus, to generate read-to-graph alignments to such assembly graphs, each homopolymer run X…X in each TELL-Seq read was collapsed into a single nucleotide X. Additionally, long dimer repeats (of length 16 and more) were compressed as described in Bankevich et al. (2022). For the homopolymer compression, we created a version of Dehomopolymerate (https://github.com/tseemann/dehomopolymerate) tool with an additional feature of dimer compression and high-throughput dataset support (https://github.com/Itolstoganov/dehomopolymerate). Assembly produced by the metaFlye was not homopolymer-compressed. Adapters were removed from TELL-Seq barcoded reads using cutadapt v 4.1 (Martin, 2011).

Representation of assembly graphs

SpLitteR was originally designed to operate on the multiplex de Bruijn graphs (mDBG) generated by the LJA assembler (Bankevich et al., 2022). However, it also supports arbitrary assembly graphs in the GFA format such as those generated by Verkko (Rautiainen et al., 2023), Flye/metaFlye (Kolmogorov et al., 2020, 2019), Shasta (Shafin et al., 2020) and other genome assembly tools. The representation of such graphs in the standard GFA format does not allow straightforward conversion to mDBG representation. For instance, the overlap length between two overlapping pairs of segments (s1, s3) and (s2, s3) should be the same for the mDBG format per the de Bruijn graph definition, while this is not the case for arbitrary assembly graphs. For that reason, we implemented an additional transformation into DBG-like graphs (GFA segments correspond to edges and unresolved repeats correspond to vertices).

Let G be an arbitrary assembly graph in the GFA format consisting of a set of segments E(G) and links L(G). We expand E(G) into a set E’(G) of segments e from E(G) and their reverse complements rc(e). Afterward, we transform G into a directed raw DBG-like graph RDG, with edges E(RDG) = E’(G). For every segment e in E’(G), we form vertices start(e) and end(e). We then merge those vertices according to the GFA links. For every GFA link (e1, e2) from L(G), we construct link edges between end(e1) and start(e2), and between end(rc(e2)) and start(rc(e1)).

We define the contraction of an edge (v, w) as merging of v and w into a single vertex u, followed by the removal of the loop-edge resulting from this merging. The DBG-like graph DG is obtained by contracting every link edge in the raw DBG-like graph RDG. For every vertex v in RDG we store GFA links L(G, v) which were contracted into v in order to retain the connectivity information from G.

For metaFlye graphs, we contract edges that were classified as repetitive by metaFlye. In the resulting contracted assembly graph CG every non-leaf vertex represents an unresolved and possibly inexact repeat from the input assembly.

Aligning barcoded reads

SpLitteR maps short reads from a linked-read library to the edges of the contracted assembly graph using the k-mer-based alignment approach originally developed for mapping Hi-C reads (Cheng et al., 2022). First, we collect unique (occurring only once) k-mers (default value k = 31) in the edge sequences of the contracted assembly graph and record their edge positions in this graph. In the metagenomic mode, we map a barcoded read-pair to an edge if both reads from the pair contain at least one unique k-mer from this edge. In the diploid mode, we relax the mapping requirement to a single k-mer from the entire read-pair, as most k-mers in the phased HiFi assembly graph are repetitive. Our analysis has shown that most 31-mers in assembly graphs constructed for a single human haplome or for a metagenome are unique, which ensures a sufficient number of unique k-mers in heterozygous regions of the assembly graph. For every edge e in the contracted graph, we store the barcode-set barcodes(e), comprising barcodes of all reads mapped to e.

Repeat resolution

In order to resolve a potential repeat, for each incoming edge e in vertex v, we aim to find the edge next(e) in the contracted assembly graph. Our base assumption is that consecutive edges e and next(e) in the genomic traversal have similar (overlapping) barcode-sets, as many fragments that contain a suffix of e also contain a prefix of next(e).

Given an in-edge e into a vertex v and an out-edge e’ from this vertex, we define overlap(e, e’) as the size of the intersection of barcode-sets barcodes(e) and barcodes(e’). For an incoming edge e into a branching vertex v, we define overlap1(e) (overlap2(e), respectively) as the largest (second largest) among all values of overlap(e,e’) among all outgoing e’ edges from v. We refer to an out-edge e’ with the largest value of overlap(e,e’) as e* (ties are broken arbitrarily).

The candidate edge next(e) for each in-edge e should satisfy two conditions. First, e and next(e) should have at least abs_thr shared barcodes (two by default). Second, the number of shared barcodes between e and next(e) should be larger than the second largest number of shared barcodes between e and out-edge of v by a relative threshold rel_thr (two by default). Specifically, we select the edge e* if overlap1(e) ≥ rel_thr * overlap2(e) and overlap1(e) ≥ abs_thr. If these conditions hold, we say that the edge-pair (e, e*) is a candidate link for vertex v. If for every edge e incident to v, |barcodes(e)| < abs_thr, we say that vertex is uncovered.

Assembly graph simplification and scaffolding

We say that a vertex v is partially resolved if candidate links constitute a non-empty matching M between in-edges and out-edges of v, and completely resolved if this matching is a perfect matching. If v is not completely resolved, partially resolved, or uncovered, we say that v is ambiguous.

For LJA assembly graphs, we perform a splitting procedure for every completely or partially resolved vertex v in order to simplify the assembly graph for possible later scaffolding. In the case of the diploid assembly graph, partially or completely resolved vertices with exactly two incoming and two outgoing edges correspond to haplotypes phased by the repeat resolution procedure. For every completely or partially resolved vertex, the set of candidate links (e, next(e)) comprises a matching. For every candidate link (e, next(e)) in this matching we create a new vertex ve, such that e is the only in-edge into ve, and next(e) is the only out-edge out of ve. If v is completely resolved, we then remove it from the graph. After performing the splitting procedure for every completely or partially resolved vertex, we condense the non-branching paths Paths in the contracted graph CG. The resulting graph is outputted in the GFA format.

For all other assembly graphs, the non-branching paths from Paths are outputted as scaffolds without changing the original assembly graph G. For every pair of scaffolded edges in CG (e, next(e)), if there is a unique path in G from end(e) to start(next(e)), the sequence of this path is inserted between e and next(e) in the resulting scaffold. Otherwise, a sequence of N characters of length DistanceG (e1, e2) is inserted between e and next(e), where DistanceG (e, next(e)) is the distance in G from end(e) to start(next(e)).

Results

We benchmarked SpLitteR on three different datasets.

The HUMAN dataset (Chen et al., 2020) was obtained from a diploid human HG002 genome that was recently assembled from HiFi reads (Rautiainen et al., 2023). The HUMAN dataset includes a TELL-Seq library which contains ~994 million barcoded TELL-Seq reads and a HiFi read-set from HG002. Since both TELL-Seq (Chen et al., 2020) and HiFi technologies (Wenger et al., 2019) emerged only 3 years ago, there are currently very few datasets that include both HiFi and TELL-Seq reads. We thus generated additional TELL-Seq datasets described below.

The HUMAN+ dataset includes two additional TELL-Seq libraries which contain an additional ~4,585 million barcoded TELL-Seq reads.

The SHEEP dataset includes a TELL-Seq library containing ~1,004 million barcoded reads and a HiFi library from a sheep fecal metagenome. Table 1 provides information about these datasets, such as approximate fragment length. The Data Preparation section specifies the details of the TELL-Seq library preparation.

Table 1 Information about TELL-Seq datasets.

Mean fragment length was evaluated based on the T2T reference for the HUMAN dataset, and the metaFlye (Nurk et al., 2020) assembly for the SHEEP dataset. A fragment is defined as the set of multiple paired-end reads with the same barcode aligned to the same long (>500 kbp) scaffold. Mean fragment length is the mean distance from the start of the leftmost aligned read in a fragment to the end of the rightmost aligned read in a fragment.

Dataset	Number of TELL-Seq reads	Genome coverage	Mean fragment length	
HUMAN	993,847,904	47×	34,317	
HUMAN+	5,579,154,072	267×	36,211	
SHEEP	1,005,083,124	N/A	27,719	

SpLitteR (version 0.1) was benchmarked against ARKS 1.2.4 (Coombe et al., 2018) and SLR-superscaffolder 0.9.1 (Guo et al., 2021) on the HUMAN, HUMAN+, and SHEEP datasets. We were not able to run Physlr (Afshinfard et al., 2022) on TELL-Seq data, while Architect (Kuleshov, Snyder & Batzoglou, 2016) was not able to finalize the scaffolding step after 14 days of runtime. We used LJA v0.2 (Bankevich et al., 2022) to generate the assembly graph (multiplex de Bruijn graph) from HiFi reads in the HUMAN and HUMAN+ datasets, and metaFlye (v.2.9) (Kolmogorov et al., 2020) to generate the assembly graph for the SHEEP dataset. Assemblies for both datasets were further scaffolded using SpLitteR, ARKS, and SLR-superscaffolder. We used QUAST-LG (Mikheenko et al., 2018) to compute various metrics of the resulting assemblies (NGA50 values, the largest alignment, etc.) with the homopolymer-compressed T2T HG002 assembly as the reference (Rautiainen et al., 2023) for the HUMAN and HUMAN+ datasets.

HUMAN dataset benchmark

We benchmarked SpLitteR, ARKS 1.2.4 (Coombe et al., 2018), and SLR-superscaffolder 0.9.1 (Guo et al., 2021) on the HUMAN dataset. In the case of SpLitteR, we benchmarked an assembly formed by sequences of homopolymer-compressed edges in the simplified assembly graph generated by the SpLitteR. In the cases of ARKS and SLR-superscaffolder, we benchmarked their assemblies formed by scaffolds of the sequences of the LJA-generated edges.

We used QUAST-LG (Mikheenko et al., 2018) to compute various metrics of the resulting assemblies with the homopolymer-compressed T2T HG002 assembly as the reference (Rautiainen et al., 2023). Table 2 illustrates that SpLitteR resulted in the largest NGA50 and NGA25 metrics for the HUMAN dataset. Specifically, NGA50 values are 301, 303, 301, and 461 kb for LJA (input graph), ARKS, SLR-superscaffolder, and SpLitteR, respectively. For the HUMAN+ dataset, the LJA assembly scaffolded with SpLitteR resulted in a 479 kb NGA50 value. Reduced total length for SpLitteR is explained by glueing together edges adjacent to resolved vertices, as the length of the vertex was included in the total length in the original LJA assembly for both in-edge and out-edge. The vertex length in the LJA graph that was used for benchmarking can reach 40 kbp. Since ARKS is using non-unique k-mers for its barcode-to-contig assignment procedure, pairs of consecutive edges originating from the same haplotype and from different haplotypes have roughly the same number of shared barcodes, which prevents accurate phasing. This might explain roughly the same NGA50 metric and higher number of misassemblies for ARKS scaffolding compared to baseline LJA assembly. SLR-superscaffolder was not able to locate unique contigs in the assembly, and thus produces the assembly identical to LJA.

Table 2 QUAST results for the HUMAN dataset.

LJA denotes the baseline LJA assembly, while the other column names correspond to a scaffolding tool applied to the baseline assembly. The best value for each row is indicated in bold.

Tool	LJA	LJA + ARKS	LJA + SLR-super scaffolder	LJA + SpLitteR	LJA + SpLitter (HUMAN+)	
# contigs (>= 0 bp)	41,407	41,092	41,407	29,587	26,912	
# contigs (>= 50 kbp)	27,508	27,315	27,508	19,946	19,049	
Total length (>= 50 kbp)	5.07 Gbp	5.07 Gbp	5.07 Gbp	4.52 Gbp	4.54 Gbp	
Largest alignment	21.6 Mbp	35 Mbp	21.6 Mbp	21.6 Mbp	21.6 Mbp	
NGA50	301,278	302,541	301,278	469,474	478,850	
NGA25	576,006	581,836	576,006	832,480	845,937	
# misassemblies	114	227	114	121	120	

ARKS generated the longest misassembly-free scaffold of length 35 Mbp (as compared to 21.6 Mbp for SpLitteR). The largest ARKS scaffold comprises three long edges of the LJA assembly graph aligned to human chromosome X. These edges (Fig. 2, blue edges) are divided by two bubbles of length approximately 80 kbp (Fig. 2, red edges). Closer investigation revealed that the central blue edge represents a graph construction artifact, since two halves of the edge were aligned by QUAST-LG to different haplotypes. As a result, the repeat shown at Fig. 2 could not be resolved by SpLitteR, as it does not contain any branching vertices incident to the central blue edge (SpLitteR only attempts to resolve repeats corresponding to branching vertices, i.e., vertices with both indegree and outdegree exceeding 1).

Figure 2 Bandage-NG plot of the ARKS largest scaffold.

Three edges comprising the largest ARKS scaffold are shown in blue. Bulge edges are shown in red. The absence of branching vertices makes it impossible for SpLitteR to resolve this component.

Repeat classification results

SpLitteR repeat resolution algorithm processes every repeat vertex with at least two in-edges and two out-edges in the assembly graph individually. We classify repeat vertices based on the results of the repeat resolution procedure. Given a repeat vertex v, we say that a pair (in_edge, out_edge) of in- and out-edges of v is connected, if out_edge follows in_edge in the genomic path according to the repeat resolution procedure. Repeat vertex v is partially resolved, if there is at least one connected pair of edges, completely resolved if all in- and out-edges of v belong to a connected pair, and uncovered if there is no in-edge and out-edge pair of edges that share at least two barcodes. If v is not completely resolved, partially resolved, or uncovered, we say that v is ambiguous. In the case of a diploid assembly, completely and partially resolved vertices correspond to regions of the assembly that were phased. For the partially resolved vertices, only one of the haplotypes was recovered using the barcode information, while the other was recovered by the process of elimination. Repeat vertex classification is described in more detail in “Repeat resolution”.

Figure 3 shows the number of completely resolved, partially resolved, ambiguous, and uncovered vertices for the HUMAN dataset depending on the length of the repeat vertex. The total number of completely resolved, partially resolved, ambiguous, and uncovered vertices for the HUMAN, HUMAN+, and SHEEP datasets is shown in Table 3. The relatively high number of ambiguous vertices in the SHEEP dataset can be explained by higher mean in- and out-degrees of vertices in the contracted metaFlye assembly graph.

Figure 3 Information about repeat resolution for the HUMAN dataset.

The x-axis shows the length of the vertex (approximate length of a repeat) in the LJA assembly graph. Barplots show the number of completely resolved (blue), partially resolved (yellow), uncovered (black), and ambiguous (red) vertices identified by the SpLitteR repeat resolution procedure. The bin width in the histogram is 2,500 bp.

Table 3 Repeat resolution statistics for the HUMAN, HUMAN+, and SHEEP datasets.

Dataset	# Completely resolved vertices	# Partially resolved vertices	# Uncovered vertices	# Ambiguous vertices	
HUMAN	7,202	2,853	6,241	547	
HUMAN+	10,681	2,405	3,526	231	
SHEEP	144	424	517	421	

Trio-binning validation

We additionally used a trio-binning tool LJATrio (Antipov, Bankevich & Bankevich, 2022), which employs the parental mother/father Illumina short reads to validate the repeat resolution procedure for a diploid dataset from a child.

LJATrio uses trio information to classify edges of the multiplex de Bruijn graph (constructed from the HiFi reads from the child dataset) into maternal, paternal, or undefined. We applied LJATrio to the mother-father-child dataset where the child corresponds to the HG002 genome and classified all edges of the corresponding contracted assembly graph into maternal, paternal, or undefined. For each vertex v, we classify its resolved in- and out-edge links as correct if both edges of the link are marked as either paternal or maternal, incorrect if one edge is marked as paternal, and the other as maternal, and unbinned otherwise. Vertex v is then classified as correct if all of its resolved links are either true or unbinned, unbinned if all resolved links are unbinned, or incorrect otherwise. In the case of diploid assembly, incorrect vertices correspond to switch errors in a diploid assembly, while unbinned vertices correspond to the unphased part of the assembly. Figure 4 shows the number of correct, incorrect, and unbinned vertices for HUMAN for completely- and partially-resolved vertices in the LJA assembly graph.

Figure 4 LJATrio validation results.

The LJATrio binning results for completely (left) and partially (right) SpLitter-resolved vertices for the HUMAN dataset. The x-axis shows the vertex length in the LJA assembly graph. The bin width in this histogram is 2,500 bp.

Coverage effects on the repeat resolution

For the HUMAN dataset, out of 6,788 branching vertices that were neither completely nor partially resolved 6,241 are uncovered, i.e., none of the in-edge and out-edge pairs share at least abs_thr barcodes. In order to analyze, how linked-read coverage affects the outcome of the SpLitteR repeat resolution procedure, we downsampled the larger HUMAN+ dataset (which in total contains ~5,579 million barcoded TELL-Seq reads) to 10%, 20%, …, 80%, and 90% of all barcodes. As shown in Fig. 5, the number of completely resolved vertices rapidly increases, while the number of uncovered vertices decreases with the increase in coverage. However, the rate of this increase slows down after 80% of barcodes are utilized, suggesting that a further increase in coverage is unlikely to significantly improve the assembly quality.

Figure 5 Repeat resolution results for downsampled HUMAN+ dataset.

The x-axis shows the selected percentage of the HUMAN+ dataset barcodes. Barplots show the number of completely resolved (blue), partially resolved (yellow), uncovered (black), and ambiguous (red) vertices identified by the SpLitteR repeat resolution procedure.

SHEEP dataset benchmark

Below we describe benchmarking of SpLitteR, ARKS 1.2.4 (Coombe et al., 2018), and SLR-superscaffolder 0.9.1 (Guo et al., 2021) on the SHEEP dataset. Unlike the HUMAN dataset which was assembled using LJA, the SHEEP dataset was assembled using metaFlye v 2.9-b1768 (Kolmogorov et al., 2020) since LJA was not designed for metagenomic assemblies. In the case of SpLitteR, we benchmarked an assembly formed by sequences of edges in the simplified assembly graph generated by the SpLitteR. In the cases of ARKS and SLR-superscaffolder, we benchmarked their assemblies based on the scaffolds provided by metaFlye.

We used QUAST-LG (Mikheenko et al., 2018) to compute various reference-free metrics of the resulting assemblies. Table 4 illustrates that all assemblers have similar contiguity with ARKS resulting in the largest NG50, NG25, and auNG metrics for the SHEEP dataset.

Table 4 QUAST results for the SHEEP dataset.

The metaFlye (edges) and metaFlye (scaffolds) denote the metaFlye assembly graph edges and the final metaFlye scaffolds, respectively. The best value for each row is indicated in bold

Tool	metaFlye (edges)	metaFlye (scaffolds)	metaFlye + ARKS	metaFlye + SLR-superscaffolder	metaFlye (edges) +
SpLitteR	
# contigs (>= 0 bp)	132,288	104,107	103,321	104,107	130,974	
# contigs (>= 50 kbp)	39,661	40,226	39,851	40,266	39,595	
Total length
(>= 50 kbp)	6.21 Gbp	6.31 Gbp	6.32 Gbp	6.31 Gbp	6.22 Gbp	
Largest contig	5,897,638	5,953,377	5,988,140	5,953,377	5,897,528	
NG50	115,660	119,205	121,777	119,205	116,285	
NG25	388,998	393,824	404,963	393,824	393,824	
auNG	396,787	399,346	413,319	399,346	401,817	

For the SHEEP dataset, SLR-superscaffolder and SpLitteR scaffolding did not result in any increase in contiguity compared to the initial metaFlye assembly, while ARKS result in a minor increase. Since ARKS and SLR-superscaffolder have very high RAM requirements, we only report SpLitteR results on the high-coverage HUMAN+ dataset.

The suboptimal results demonstrated by SpLitteR assembly compared to ARKS and the base metaFlye assembly stem from the negligible amount of information provided by the assembly graph structure. The metaFlye assembly graph of the SHEEP dataset contains only 923 branching vertices, of which 194 were completely resolved and 433 were partially resolved by SpLitteR. Despite the relative effectiveness of the SpLitteR repeat resolution procedure (more than half the branching vertices were at least partially resolved), graph-agnostic scaffolding performed by ARKS yields more contiguous assembly. The results for the SHEEP dataset demonstrate that the effectiveness of SpLitteR’s scaffolding highly depends on the choice of the baseline assembly graph.

Discussion

Since linked-read scaffolders, such as SLR-superscaffolder and ARKS, do not utilize the assembly graph information, they have limited applicability to long-read assemblies due to the increased unresolved repeat length compared to short read assemblies. In addition, SLR-superscaffolder utilizes input .bam file to assign barcodes to contigs, while ARKS uses non-unique kmers for the same purpose. In the case of highly repetitive assembly graphs, e.g., constructed from diploid genomes or strain-rich metagenomes, resulting barcode assignments turn out to be inaccurate. Producing inaccurate barcode-edge assignments and ignoring connections between assembly graph edges makes it difficult for both ARKS and SLR-superscaffolder to improve upon baseline LJA assembly in the HUMAN dataset. On the SHEEP dataset, the baseline assembly graph has higher connectivity and thus provides less information that can be used for scaffolding, while the assembly is less repetitive. As a result, ARKS is able to outperform baseline assembly and other scaffolders. Both ARKS and SLR-superscaffolder output scaffolds instead of an assembly graph, which makes it harder to use TELL-Seq in combination with other supplementary sequencing technologies.

SpLitteR uses the assembly graph and employs unique k-mer mapping to overcome these shortcomings. For high quality assembly graphs, even the simple SpLitteR repeat resolution algorithm resolves 94.6% repeats in the HUMAN dataset that are bridged by at least two TELL-Seq fragments. However, for the SHEEP metagenome assembly graph with less clear repeat structure, SpLitteR results in less contiguous assembly than ARKS. While SpLitteR is in theory able to take as an input assembly graphs other than metaFlye and LJA, other assemblers which support GFA format are not yet supported. It should also be noted that for the highly repetitive assemblies consisting of long exact repeats with relatively few SNPs, TELL-Seq short read coverage should be quadratic with respect to ultralong read coverage, since two reads in the same barcode should cover two SNP positions in order to provide information, while a single ultralong read is able to cover consecutive SNPs. Despite this limited coverage scalability compared to ultralong reads, using TELL-Seq human genome dataset with 25× coverage was enough to resolve 62% of repeats unresolved by HiFi assembly.

Conclusions

We developed the SpLitteR tool for scaffolding and assembly graph phasing using linked-reads. Our benchmarking demonstrated that it significantly increases the assembly contiguity compared to the previously developed HiFi assemblers and linked-read scaffolders. We thus argue that linked-reads have the potential to become an inexpensive supplementary technology for generating more contiguous assemblies of large genomes from the initial HiFi assemblies, in line with ONT and Hi-C reads, which were used by the T2T consortium to assemble the first complete human genomes (Rautiainen et al., 2023; Nurk et al., 2022). Since the assembly graph simplification procedure in SpLitteR yields longer contigs as compared to the initial HiFi-based assembly, SpLitteR can be integrated as a preprocessing step in the assembly pipeline with other tools that employ supplementary sequencing technologies, such as Hi-C (Cheng et al., 2021) and Strand-seq (Porubsky et al., 2021).

Ivan Tolstoganov and Anton Korobeynikov are grateful to Saint Petersburg State University for providing the computational resources for the experiments that were performed on a high performance computational server.

Additional Information and Declarations

Competing Interests

Author Contributions

DNA Deposition

Data Availability

Zhoutao Chen declares competing financial interests in the form of stock ownership, patent application, or employment through Universal Sequencing Technology Corporation. Other authors declare that they have no competing interests.

Ivan Tolstoganov conceived and designed the experiments, performed the experiments, analyzed the data, prepared figures and/or tables, authored or reviewed drafts of the article, and approved the final draft.

Zhoutao Chen analyzed the data, authored or reviewed drafts of the article, data Preparation, and approved the final draft.

Pavel Pevzner conceived and designed the experiments, authored or reviewed drafts of the article, and approved the final draft.

Anton Korobeynikov conceived and designed the experiments, prepared figures and/or tables, authored or reviewed drafts of the article, and approved the final draft.

The following information was supplied regarding the deposition of DNA sequences:

The sequencing reads for the HUMAN dataset are available at NCBI BioProject: SRX7264481. The remaining reads for the HUMAN+ and SHEEP datasets generated in this study are available at NCBI BioProject: PRJNA956112.

The following information was supplied regarding data availability:

SpLitteR is implemented in C++ as a part of the freely available under GPL license SPAdes package and is available at GitHub and Zenodo:

- https://github.com/ablab/spades/releases/tag/splitter-preprint

- Anton Korobeynikov, Sergey Nurk, Dmitry Antipov, Andrey Prjibelski, Mikhail Dvorkin, AntonBankevich, Alexander Shlemov, Nikolay Vyahhi, Alexey Gurevich, Alexander Sirotkin, Yulia Gorshkova, Mariya Davydova, Olga Kunyavskaya, Sergey Nikolenko, Alex, Alex Davydow, valery-l, Alexander S. Kulikov, Anton Kleshchin, … Anton Garder. (2024). ablab/spades: Release v4.0.0 (v4.0.0). Zenodo. https://doi.org/10.5281/zenodo.11465940

Baseline LJA assembly and trio binning results for the HUMAN+ dataset are available at Zenodo: Tolstoganov, I. (2024). Assembly graphs and reference Verkko assembly for HG002 dataset [Data set]. Zenodo. https://doi.org/10.5281/zenodo.11661572.

The baseline metaFlye assembly for the SHEEP dataset is available at Zenodo: Tolstoganov, I. (2024). Assembly graphs and reference Verkko assembly for HG002 dataset [Data set]. Zenodo. https://doi.org/10.5281/zenodo.11661572.

The Spades assembler license is available at GitHub:

https://github.com/ablab/spades/blob/ec4371ce9207a89a6d71c3e7256825f1fa83d6c6/assembler/LICENSE.

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
