# Peer review of "SpLitteR: diploid genome assembly using TELL-Seq linked-reads and assembly graphs"

_PeerJ, doi:10.7717/peerj.18050_

## Round 0.1 · original submission · Minor Revisions

Please address issues raised by the reviewers, in particular a discussion of haplotypes and their phasing in relation to the results, and the clarifications in the method description and exposition as requested.

With respect to additional references, I refer to PeerJ policy. I do think that adding a paragraph or two to the beginning of the introduction, placing the work in a bigger context within genome biology (and not focussed on tools), could potentially very nicely round out the manuscript and make it more accessible for readers who are not method developers.

·

Basic reporting

Tolstoganov et al report SpLitteR, a tool to resolve assembly graphs using linked reads. Overall the work is interesting, novel, and well presented. The main shortcoming that I would like to see addressed is the absence of specific considerations regarding haplotypes and their phasing, even though it's actually mentioned as one of the goals/results in the abstract. Yet this is a critical aspect in diploid genome assembly. It is not clear how haplotypes are impacted (for instance in Fig1), apparently they were not directly evaluated or results are not reported/discussed. It is also unclear how the algorithm could be integrated in tools that perform haplotype-resolution (presumably at a later stage).
Minor:
"Afterward, long fragments within the containers are sheared into shorter fragments and sequenced." > the barcoding step should also be mentioned
"However, even though linked-reads result in more contiguous assemblies than assemblies based on non-linked short reads, all these tools generate rather fragmented assemblies of large genomes and metagenomes." > references needed, e.g. https://www.nature.com/articles/s41586-021-03451-0
Isn't CTGA miscolored in figure 1 (shouldn't it be ACT or ACTG)
It isn't clear if homopolymer compression is a feature of the tool or needs to be performed separately (if so, it is not specified how)
line 132: bold

Experimental design

In line with the scope of the journal. Rigorous work.

Validity of the findings

Impact and novelty are significant. Support data well presented, conclusions well-stated

·

Basic reporting

Comments:

- The abstract and introduction are well-crafted, providing a clear overview.
- The manuscript acknowledges limitations effectively, notably the challenges with HiFi-based assemblies in repeat-rich areas, where repeats can extend beyond the length of HiFi reads, impacting the continuity of diploid genome assemblies due to shared long identical regions between two haplomes. Tell-seq data, though leading to more contiguous or fragmented assemblies when used alone, offers a solution. Please cite review article: https://genomebiology.biomedcentral.com/articles/10.1186/s13059-021-02328-9

- The process of aligning barcoded reads, achieving uniqueness with a k-mer length of k=31 for edge sequences, and its relation to the underlying graph theory is well explained, though more integration with graph theoretical principles could enhance understanding. Please refer article for readers: https://www.nature.com/articles/s41467-023-36689-5.

- How well does this method work in highly polymorphic regions?
- The results section is well-supported with detailed information on read length, number of reads, coverage, and post-assembly data quality presented in informative tables. The use and evaluation of tools for graph assembly are thoroughly described.
- The method section's introduction (lines 83-87) disrupts the flow, with suggestions to align the explanation of branching more closely with Figure 1 or to integrate it into lines 89-99.
- References to the “manufacturer’s user guide” are missing in the data preparation section (lines 114, 121).
- Lack of comparison between results from multiplex de Bruijn graphs (mDBG) and other types of assembly graphs derived from GFAs.
- Unclear descriptions are noted in several areas: conversion from a standard GFA to mDBG encoding (lines 136-138), and the notation of graph edges and vertices (lines 140-145), recommending adherence to standard notations as in "Introduction to Algorithms" (CLRS). The description of rel_thr and abs_thr parameters (lines 182-183) is also vague, with only values provided.

Experimental design

The experimental design seems appropriate, well-defined, relevant, and meaningful.

Validity of the findings

The findings are valid and novel.

---

## Round 0.2 · accepted · Accept

Thank you for addressing comments/suggestions from the previous academic editor. The manuscript is now accepted for publication in PeerJ.

·

Basic reporting

The authors addressed all my concerns and I am happy with the revised manuscript.

Experimental design

Already answered previously. All good.

Validity of the findings

Already answered previously. All good.

·

Basic reporting

The authors have addressed my comments.

Experimental design

The authors have addressed my comments.

Validity of the findings

The authors have addressed my comments.